# Candidate Gene Identification and Transcriptome Analysis of Tomato *male sterile*-*30* and Functional Marker Development for *ms*-*30* and Its Alleles, *ms*-*33*, *7B*-*1*, and *stamenless*-*2*

**DOI:** 10.3390/ijms25063331

**Published:** 2024-03-15

**Authors:** Kai Wei, Xin Li, Xue Cao, Shanshan Li, Li Zhang, Feifei Lu, Chang Liu, Yanmei Guo, Lei Liu, Can Zhu, Yongchen Du, Junming Li, Wencai Yang, Zejun Huang, Xiaoxuan Wang

**Affiliations:** 1State Key Laboratory of Vegetable Biobreeding, Institute of Vegetables and Flowers, Chinese Academy of Agricultural Sciences, Beijing 100081, China; 82101209110@caas.cn (K.W.); lixin10@caas.cn (X.L.); xsunshine11@163.com (X.C.); 18801367465@163.com (S.L.); 82101219106@caas.cn (L.Z.); 15514907620@163.com (F.L.); 82101225062@caas.cn (C.L.); guoyanmei@caas.cn (Y.G.); liulei02@caas.cn (L.L.); zhucan@caas.cn (C.Z.); duyongchen@caas.cn (Y.D.); lijunming@caas.cn (J.L.); 2Department of Vegetable Science, College of Horticulture, China Agricultural University, Beijing 100193, China; yangwencai@cau.edu.cn

**Keywords:** tomato, male sterility, *Tomato PISTILLATA*, functional molecular marker, gene editing, transcriptome analysis

## Abstract

Male sterility is a valuable trait for hybrid seed production in tomato (*Solanum lycopersicum*). The mutants *male sterile*-*30* (*ms*-*30*) and *ms*-*33* of tomato exhibit twisted stamens, exposed stigmas, and complete male sterility, thus holding potential for application in hybrid seed production. In this study, the *ms*-*30* and *ms*-*33* loci were fine-mapped to 53.3 kb and 111.2 kb intervals, respectively. *Tomato PISTILLATA* (*TPI*, syn. *SlGLO2*), a B-class MADS-box transcription factor gene, was identified as the most likely candidate gene for both loci. *TPI* is also the candidate gene of tomato male sterile mutant *7B*-*1* and *sl*-*2*. Allelism tests revealed that *ms*-*30*, *ms*-*33*, *7B*-*1*, and *sl*-*2* were allelic. Sequencing analysis showed sequence alterations in the *TPI* gene in all these mutants, with *ms*-*30* exhibiting a transversion (G to T) that resulted in a missense mutation (S to I); *ms*-*33* showing a transition (A to T) that led to alternative splicing, resulting in a loss of 46 amino acids in protein; and *7B*-*1* and *sl*-*2* mutants showing the insertion of an approximately 4.8 kb retrotransposon. On the basis of these sequence alterations, a Kompetitive Allele Specific PCR marker, a sequencing marker, and an Insertion/Deletion marker were developed. Phenotypic analysis of the *TPI* gene-edited mutants and allelism tests indicated that the gene *TPI* is responsible for *ms*-*30* and its alleles. Transcriptome analysis of *ms*-*30* and quantitative RT-PCR revealed some differentially expressed genes associated with stamen and carpel development. These findings will aid in the marker-assisted selection for *ms*-*30* and its alleles in tomato breeding and support the functional analysis of the *TPI* gene.

## 1. Introduction

Tomato (*Solanum lycopersicum*) is among the most crucial vegetables globally. According to the Food and Agriculture Organization of the United Nations statistical database 2021, approximately 189 million tons of fresh tomatoes are produced worldwide (https://www.fao.org/faostat/zh/#data/QCL). The widespread adoption of tomato F_1_ hybrids can be attributed to their consistently higher yields, improved quality, and enhanced resistance to diseases compared with open-pollinated varieties [1,2]. However, the traditional method of producing tomato hybrid seeds involves labor-intensive processes, such as manual emasculation and hand pollination, leading to increased costs and the risk of self-pollination impurities [3,4]. To address these challenges, researchers have explored the suitability of male sterile plants as female progenitors in the production of tomato hybrid seeds [4,5]. Different genes controlling male sterility have been examined, and their applications in plant breeding have been evaluated.

Since the first report of a male sterile mutant in tomatoes in 1915, over 55 male sterile alleles have been identified and categorized into three types: sporogenous, functional, and structural sterility [5,6]. Most tomato male sterile mutants are of the sporogenous type [7], with only a few having been cloned, such as *SlMS10* and *SlMS32*.Both of these encode a basic helix–loop–helix (bHLH) transcription factor [8,9,10] and are homologous to *DYT1* and *bHLH10/89/90* in *Arabidopsis*, respectively [8,10]. DYT1 and bHLH10/89/90 are part of the DYT1-TDF1-AMS-MS188-MS1 pathway that regulates tapetum development and pollen formation in *Arabidopsis* [11]. *Positional sterile* (*ps*) is the first reported mutant of functional male sterility in tomato [12]. It produces normal pollen; however, its pollen grains remain enclosed within the anther locule due to non-splitting anthers [13]. The second reported mutant of this type is *ps*-*2*. *PS*-*2* encodes a polygalacturonase gene (*PG*) [14,15], and it has been used for tomato hybrid seed production in Eastern Europe [16]. Moreover, functional male sterility can result from exserted stigma. In the *exserted stigma* (*ex*) mutant, the anthers are normal and capable of producing and releasing viable pollens, but the excessively long style prevents effective pollination [17]. Other mutants of this type include *Style2.1*, *Stigma Exertion 3.1* (*SE3.1*), and *SlLst* (*Long styles*) [18,19,20]. *Stamenless* (*sl*), the first structural male sterile mutant reported in tomato, exhibited the complete transformation of stamens into carpels and the partial transformation of petals into sepals [21]. *Tomato APETALA3* (*TAP3*) may be responsible for the *sl* trait [22,23]. In the *sl*-*2* mutant, petals appear almost typical, but the stamens are twisted and distorted, bearing naked ovules [24]. The male sterile mutant *7B*-*1* displays a high degree of phenotypic similarity with *sl*-*2* and an allelism test confirmed that *7B*-*1* and *sl*-*2* are allelic [25]. Genetic mapping and molecular analysis indicated that the B-class MADS-box gene *Tomato PISTILLATA* (*TPI*, syn. *SlGLO2*) is a candidate gene of *7B*-*1* and *sl*-*2*; however, the specific sequence variations of *TPI* in *7B*-*1* and *sl*-*2* mutants remain elusive [25]. Tomato *ms*-*15* and its alleles, *ms*-*15^26^* and *ms*-*15^47^*, producing flowers with homeotic conversion of stamens into carpels, have been fine-mapped to a 44.6 kb interval that contains the *TM6* gene [26]. A recent study revealed that *TM6* maintains the balance of alternative splicing of MADS-box genes involved in flower development [27].

Each type of male sterility in tomato presents both advantages and disadvantages for use in hybrid seed production. The lines *ms10* and *ms32* are considered beneficial for hybrid seed production [28], because they exhibit exserted stigma (accessible for pollination without emasculation), complete sterility, and stable expression of sterility regardless of environmental conditions. However, the application of these two sterile lines has some major drawbacks, such as the variation in the percentage of flowers with accessible stigmas depending on the environment and genotype [4]. The *exserted stigma* (*ex*) sterility type was initially considered advantageous for hybrid seed production due to the easy maintenance of sterile lines, absence of a need for stamen emasculation, and high hybrid seed yield. However, it presents the challenge of managing two polygenic traits (anther and style length) and the variability of phenotypic expression under different environmental conditions. *ps-2* has disadvantages due to the risk of undesirable self-pollination and the necessity for stamen emasculation [4,16]. Although *7B-1* can be used for hybrid seed production [29], the absence of a functional codominant marker for *7B-1* reduces the efficiency of backcross transfer. These deficiencies in male sterile lines for tomato hybrid seed production necessitate further exploration of male sterility genes and the development of codominant molecular markers for marker-assisted selection (MAS).

Tomato male sterile mutants *ms*-*30* and *ms*-*33* feature shorter, twisted stamens and exerted stigmas, facilitating direct pollination without the need for emasculation [30,31]. Among these mutants, *ms*-*33* appears highly promising as a female parent in the production of tomato hybrid seeds [32]. Linkage analysis revealed that the *ms*-*33* locus is located on chromosome 6 [33]. The fertility of *ms*-*33* can be partially or completely restored with the application of exogenous GA3, resulting in seeds that produce 100% male sterile plants [7,34]. Given these characteristics, *ms*-*30* and *ms*-*33* can be valuable for hybrid seed production. However, the specific genes responsible for *ms*-*30* and *ms*-*33* sterility remain unknown, and there are currently no codominant functional markers available for selecting these mutants.

The objective of this study was to characterize the genes *MS-30* and *MS-33*, along with their sequence variations in wild-type plants and the male sterile mutants *ms-30* and *ms-33*. Subsequently, the study aimed to develop codominant functional markers for MAS utilizing these sequence variations, and to perform an initial investigation into the role of the *MS-30* gene in stamen development through RNA-seq analysis.

## 2. Results

### 2.1. Phenotypes of the ms-30 and ms-33 Mutants

The flowers of WT-30 (the wild-type corresponding to *ms*-*30*) and WT-33 exhibited normal architecture, in contrast to the stamens of *ms*-*30* and *ms*-*33*, which displayed distinctive features, such as twisted stamens, lateral separation, reduced size, and pale color, resulting in an exerted stigma (Figure 1 and Appendix A). Additionally, these distorted stamens were observed to transform into carpel-like structures, including carpelloid structures (CS), naked external ovules on the adaxial surface (EO), and complete transformation into carpels (TC). Notably, the occurrence of EO was more frequent than that of CS and TC in both *ms*-*30* and *ms*-*33* (Appendix A). Due to these alterations, neither mutant was capable of producing pollens (Appendix A), and both *ms*-*30* and *ms*-*33* were unable to set fruit when subjected to self-pollination using an electric vibrator. However, manual pollination with pollens from wild-type plants resulted in a successful fruit set (Appendix A).

### 2.2. Genetic Analysis of ms-30 and ms-33 Loci

The F_1_ plants resulting from crosses between *ms*-*30* and LA1589, as well as between *ms*-*33* and LA1589, displayed complete fertility. Additionally, a preliminary analysis involved 200 plants in each of the F_2_ populations derived from these crosses. The ratio of fertile to sterile plants for both *ms*-*30* and *ms*-*33* conformed to the expected 3:1 segregation ratio, as confirmed by the chi-square test (Table 1). The observed segregation pattern and the statistical analysis suggested that the male sterility in both *ms*-*30* and *ms*-*33* is controlled by a single recessive gene.

### 2.3. Fine-Mapping of ms-30 and ms-33 Loci

For preliminary mapping of the *ms*-*30* and *ms*-*33* loci, markers located on chromosome 6 were utilized. This approach was based on prior reports indicating that *ms*-*33* is located on chromosome 6 of tomato. Furthermore, the *ms*-*30* and *ms*-*33* mutants exhibited similar phenotypes, suggesting a potential linkage (Figure 1 and Appendix A) [30,31,33,35,36]. In this preliminary mapping phase, 94 male sterile plants from each F_2_ population were genotyped, and their recombination rates were calculated. As anticipated, the findings confirmed that both *ms*-*30* and *ms*-*33* are located within the region bounded by markers HP149 and HP3661 on chromosome 6 (Figure 2A and Appendix A). To further narrow down the locations of these loci, 76 and 39 recombinants were selected from the entire F_2_ populations of *ms*-*30* and *ms*-*33*, respectively, and genotyped with additional markers. This detailed analysis allowed the *ms*-*30* locus to be narrowed down to a 53.3 kb region, delimited by the markers CP207 and CP203 (Figure 2B). Similarly, the *ms*-*33* locus was confined to an interval of 111.2 kb, defined by markers CP207 and HP3921 (Appendix A). Notably, both loci shared an identical 53.3 kb region.

### 2.4. Candidate Gene Analysis and Allelism Testing among ms-30, ms-33, and sl-2

According to the ITAG4.0 genome annotation [37], *Solyc06g059970*, which encodes *Tomato PISTILLATA* (*TPI*, synonymously referred to as *SlGLO2*) [38,39], was identified within the fine-mapped regions of both the *ms*-*30* and *ms*-*33* loci (Figure 2C and Appendix A; Table 2 and Appendix A). *TPI* plays a pivotal role in stamen development and has been identified as the candidate gene for *7B*-*1* and *sl*-*2* [25], which exhibit phenotypes similar to those of *ms*-*30* and *ms*-*33* (Figure 1 and Appendix A). To further ascertain the genetic relationships among *ms*-*30*, *ms*-*33*, and *sl*-*2*, allelism tests were performed. The outcomes of these tests confirmed that *ms*-*30*, *ms*-*33*, and *sl*-*2* are allelic, indicating that they arise from mutations in the same gene (Appendix A). Thus, *TPI* was identified as the most probable candidate gene responsible for the phenotypic traits observed in *ms*-*30* and *ms*-*33*.

### 2.5. Sequence Analysis of the TPI Gene in ms-30, ms-33, 7B-1, and sl-2

To identify sequence variations within the *TPI* gene between wild-type plants and *ms* mutants, the genomic sequence of the *TPI* gene was sequenced in WT-30, *ms-30*, WT-33, and *ms-33*. Sequencing data analysis revealed specific mutations in each mutant. In *ms-30*, a single-nucleotide polymorphism (SNP) was discovered in the first exon of the *TPI* gene (Figure 3A), leading to a missense mutation. This mutation altered a serine residue to an isoleucine within the MADS domain of the protein, impacting its function (Appendix A). In *ms-33*, a different SNP was identified in the first intron of the *TPI* gene (Figure 3B). Amplification of cDNAs from *ms-33* produced several bands, with one being smaller than the corresponding band from the wild-type (Appendix A). Further analysis indicated that this SNP might introduce an alternative 5’ splice site in *ms-33*’s gene transcript. This alternative splicing resulted in a 138 bp deletion in cDNA, leading to the protein lacking a 46 amino-acid fragment (Appendix A). 

Pucci et al. (2017) [25] reported that the primer pair LeGLO2-6F/R did not amplify the last exon and part of the 3’ UTR region of the *TPI* gene in *7B-1* and *sl-2*. To investigate sequence variation in this region, PCR amplification was conducted using the LeGLO2-6F/R primer pair with an extended elongation time. This method enabled the production of approximately 5000 bp long amplicons from both *7B-1* and *sl-2* (Appendix A). Sequencing of these amplicons showed no sequence variations. However, a fragment of 4868 bp in the amplicon was identified as a *Copia*-like retrotransposon, featuring a 398 bp long terminal repeat (LTR) sequence and a 4 bp target site duplication (TSD) sequence (Figure 3C and Appendix A). Based on the SNPs in the *TPI* gene of *ms-30* and *ms-33* and the LTR retrotransposon insertion in the *TPI* gene of *7B-1* and *sl-2*, three codominant markers were developed, namely a Kompetitive Allele Specific PCR (KASP) marker, MS30KASP; a sequence marker, MS33SEQ; and an Insertion/Deletion (InDel) marker, LeGLO2-6/SL-2TE (Figure 3D–F).

### 2.6. Targeted Knockout of the TPI Gene by CRISPR/Cas9

To confirm that the *TPI* gene confers the phenotype observed in *ms-30* and its alleles, the *TPI* gene was knocked out using the CRISPR/Cas9 system, and allelism tests were conducted between the *ms-30* and *TPI* gene-edited mutants. Two vectors were designed to disrupt the *TPI* gene, with target sites on the first and last exons for vector-1 and vector-2, respectively. This resulted in two null mutants of the *TPI* gene. The *tpicr-e1* mutant exhibited a 4 bp deletion, whereas *tpicr-e7* had a 66 bp deletion (Figure 4A). The stamens of both *tpicr-e1* and *tpicr-e7* mutants showed curling, wrinkling, greenish coloration, and were incapable of producing viable pollen (Figure 4B–J), similar to phenotypic traits observed in the *ms-30* and *ms-33* mutants. Allelism tests between *tpicr-e1*, *tpicr-e7*, and *ms-30* indicated that they were allelic (Appendix A). These findings support the conclusion that *TPI* is responsible for the observed phenotypes in *ms-30* and its alleles, including *ms-33*, *7B-1*, and *sl-2*.

### 2.7. Comparative RNA-Seq Analysis of WT-30 and ms-30 Floral Organs from the Meiotic to Tetrad Stage

To further explore the role of *TPI* in floral organ development, RNA-seq analysis was conducted on the four floral organ whorls of young flower buds from WT-30 and *ms-30* at the meiotic and tetrad stages (3–5 mm in length) [40]. RNA-Seq generated approximately 568 million paired-end reads, with each sample yielding between 20 million and 30 million reads. On average, 98% of these reads were successfully mapped to the ITAG4.0 tomato reference genome (Appendix A). Principal component analysis (PCA) results showed that biological replicates clustered together, indicating good consistency within the data (Figure 5A). Notably, the stamens of *ms-30* (ST_*ms-30*) were closely related to the carpels of WT-30 (CA_WT-30) and the carpels of *ms-30* (CA_*ms-30*) in the PCA plot. This observation aligns with the homeotic transformation of stamens into carpels observed in the *ms-30* mutant, providing molecular evidence for this phenotypic change.

Among the differentially expressed genes (DEGs) identified between WT-30 and *ms-30*, 96 were in sepals, 738 in petals, 5391 in stamens, and 131 in carpels. Specifically, within the stamens, 2238 genes were upregulated, whereas 3153 genes were downregulated (Figure 5B, Appendix A). Regarding the ABCDE model of floral organ development, the expression of the D-class gene *TAGL11* was not detected in any of the four floral organ whorls. The expression levels of other ABCDE model genes, with the exceptions of *SlGLO1* and *TAG1*, differed significantly in the stamens between WT-30 and *ms-30* (Appendix A).

Gene ontology (GO) analysis revealed that upregulated DEGs in stamens were enriched in specific biological process categories, such as cell division, regulation of transcription, DNA-templated synthesis, response to auxin, ethylene-activated signaling pathways, and response to light stimulus (Figure 5C). Conversely, DEGs downregulated in stamens were associated with categories such as response to water deprivation, response to abscisic acid, pollen sperm cell differentiation, and pollen tube growth (Figure 5D). These findings suggest that the expression of the identified DEGs is associated with key processes such as pollen development, hormonal responses, and responses to environmental stimuli.

Given the homeotic conversion of stamens into carpel-like organs observed in *ms-30*, our analysis focused on the DEGs related to stamen and carpel development. DEGs with homologs involved in stamen or carpel development in *Arabidopsis* were identified in the stamens of *ms-30*. Some of these genes have been previously reported to play roles in stamen or carpel development in tomato (Figure 6A,B, Appendix A). For example, genes associated with tomato tapetum development, such as *SlMS10* (*Solyc02g079810*), *Solyc03g113530* (an *AtTDF1* homolog), *SlAMS* (*Solyc08g062780*), *SlMS32* (*Solyc01g081100*), and *SlPHD_MS1* (*Solyc04g008420*) were downregulated in the stamens of the ms-30 mutant (Figure 6C) [8,9,10,41,42]. Conversely, genes related to tomato carpel or ovule development, including *SlCRCa* (*Solyc01g010240*), *SlCRCb* (*Solyc05g012050*), *SlINO* (*Solyc05g005240*), *LYRATE* (*Solyc05g009380*), and *SlMBP3* (*Solyc06g064840*), were upregulated in the stamens of the *ms-30* mutant (Figure 6D) [43,44,45,46]. These expression differences were validated through qRT-PCR assays (Figure 6C,D). Moreover, qRT-PCR analysis showed that these genes were also differentially expressed in the *ms-33*, *tpicr-e1*, and *tpicr-e7* mutants (Appendix A), further supporting their involvement in the observed phenotypic alterations.

## 3. Discussion

Using male sterile lines as the female parent in hybrid tomato seed production is considered a highly efficient and cost-effective method [28]. Several male sterile lines, including *ms-10*, *ms-32*, *sl*, *7B-1*, *ex*, *ps*, and *ps-2*, [4,29] have been successfully applied in tomato hybrid seed production [4,29]. However, these lines have drawbacks, such as the risk of self-pollination, the need for anther emasculation, environmental dependency of phenotypic expression, reduced hybrid seed yield, or the lack of codominant markers for MAS [4]. Therefore, identifying additional genes responsible for male sterility in tomatoes, particularly those resulting in exposed stigmas and stable, complete male sterility, is crucial. Developing codominant markers for MAS of these male sterile loci is equally important. The *ms-30* and *ms-33* mutants are promising for hybrid tomato seed production due to their complete male sterility, exerted stigmas (Figure 1), and the ability to be manually pollinated directly without the need for hand emasculation. Both *ms-30* and *ms-33* exhibit single-gene recessive inheritance, with only homozygous plants displaying the male sterile phenotype, making MAS more efficient than conventional phenotypic selection for these mutants in breeding programs [47]. The *ms-30* and *ms-33* loci have been fine-mapped, identifying the *TPI* gene as the candidate gene for both loci. These mutants were found to be allelic to *7B-1* and *sl-2* [25], which also involve *TPI*. Phenotypic analysis and allelism tests of *TPI* gene-edited mutants confirmed that *TPI* underlies the male sterile phenotype in *ms-30* and its alleles, including *ms-33*, *7B-1*, and *sl-2*. Sequence analysis identified variations in the *TPI* gene across these mutants. Based on these variations, three codominant markers—KASP marker MS30KASP, sequencing marker MS33SEQ, and InDel marker LeG-LO2-6/SL-2TE—were developed, facilitating rapid incorporation of these loci into tomato breeding. Additionally, CRISPR/Cas9-mediated targeted mutagenesis represents another approach to rapidly introduce male sterility into elite breeding lines [9,48,49,50,51,52]. The phenotypes of two *TPI* gene-edited mutants closely matched those of *ms*-*30* and its alleles, suggesting that gene editing of *TPI* could efficiently induce male sterility in elite lines, thereby reducing the costs associated with hybrid seed production.

Carpelloid stamens are occasionally observed in angiosperms, with mutations in B-class genes identified as the primary genetic cause of this phenomenon [53]. The number of B-class genes varies across different plant species; for instance, *Arabidopsis* has two B-class genes (*AP3* and *PI*), whereas tomato contains four genes (*TAP3*, *TM6*, *TPI*, and *SlGLO1*) [38,54]. This variation suggests potential functional conservation and diversification of B-class genes among plant species [54]. The role of B-class genes in stamen specification and development has been extensively studied since the 1990s [55,56,57]; however, further research is warranted. In this study, the *ms-30* and *ms-33* mutants displayed carpelloid stamens (Figure 1) and were found to be allelic to *7B-1* and *sl-2* (Appendix A). The B-class gene *TPI* was identified as the underlying gene for *ms-30* and its alleles (Figure 2 and Appendix A). Transcriptome analysis of *ms-30* was conducted to examine the function of *TPI* in stamen development. PCA of the transcriptome data showed that the clusters for stamens and carpels were closely related (Figure 5A), aligning with the observed phenotype of carpelloid stamens in *ms-30* (Figure 1). The number of DEGs was higher in stamens than in the other three floral organs (Figure 5B). Some DEGs in stamens, specifically expressed in wild-type stamens, were downregulated in the *ms-30* mutant (Figure 6A). Examples of such genes are *SlMS10* (*Solyc02g079810*), *Solyc03g113530* (*AtTDF1-like1*), *SlAMS* (*Solyc08g062780*), *SlMS32* (*Solyc01g081100*), and *SlPHD_MS1* (*Solyc04g008420*) (Figure 6C), which are key components of the DYT1-TDF1-AMS-MS188-MS1 regulatory network [8,9,10,41,42]. Conversely, DEGs that were specifically expressed in wild-type carpels and upregulated in *ms-30* stamens included *SlCRCa* (*Solyc01g010240*), *SlCRCb* (*Solyc05g012050*), *SlINO* (*Solyc05g005240*), *LYRATE* (*Solyc05g009380*), and *SlMBP3* (*Solyc06g064840*) (Figure 6D), suggesting their involvement in carpel or ovule development in tomato [43,44,45,46]. These findings suggest that *TPI* positively regulates the genes associated with stamen and pollen development while negatively influencing the genes involved in carpel and ovule development within stamens. Consequently, loss of function of the *TPI* gene leads to the low or absent expression of genes associated with stamen development and the ectopic expression of genes related to carpel and ovule development in stamens, potentially resulting in the formation of carpelloid stamens in mutants. However, further investigation is needed to elucidate the specific molecular regulatory mechanisms.

## 4. Materials and Methods

### 4.1. Plant Materials

Seeds for the wild-type (WT) plants and *ms* mutants (*ms-30*, accession number 2-455; *ms-33*, 2-511; *sl-2*, LA1801; Ailsa Craig [AC]; LA1589 and Heinz 1706) were obtained from the Tomato Genetics Resource Center (Davis, CA, USA). The male sterile plants derived from the accessions 2-455, 2-511, and LA1801 were designated as *ms-30*, *ms-33*, and *sl-2*, respectively. Similarly, the homozygous male fertile plants from 2-455 and 2-511 were identified as WT-30 and WT-33, respectively. AC was used for gene editing and as WT controls for the *tpicr-e1* and *tpicr-e7* mutants. Two F_2_ populations were created by crossing both *ms-30* and *ms-33* with the wild tomato species *Solanum pimpinellifolium* accession LA1589. These populations were cultivated in an open field in Shunyi District, Beijing, China, during the spring and summer of 2016, comprising 949 and 861 plants, respectively.

To confirm the allelism among *ms-30*, *ms-33*, and *sl-2*, the flowers of *sl-2* were pollinated using pollens from the heterozygous *ms-30* or *ms-33* plants. Additionally, to determine whether the *TPI* gene is responsible for conferring the *ms-30* phenotype, flowers of *ms-30* were pollinated with pollens from the heterozygous *TPI* gene-edited mutants. Conversely, male sterile flowers of *TPI* gene-edited mutants were pollinated with pollens from the heterozygous *ms-30* plants. The progeny from these crosses were grown in a greenhouse located in Haidian District, Beijing, China, during the springs and summers of 2019, 2020, and 2023.

### 4.2. Phenotypic Analysis

For phenotypic and expression analyses, WT-30, *ms-30*, WT-33, *ms-33*, and AC plants, as well as gene-edited mutants, were grown in a greenhouse in Haidian District, Beijing, China, across the springs and summers of 2019, 2020, 2021, and 2023. The flower morphology of both WT and *ms* mutants was examined at the anthesis stage using the previously described methods [25]. The entire flower was photographed by a camera (Canon EOS 70D, Canon Inc., Tokyo, Japan). The anther cones and dissected stamens were observed by a stereomicroscope (Carl Zeiss MicroImaging GmbH, Gottingen, Germany). Furthermore, referring to the procedures outlined by Cao et al., whole plants were photographed and pollen viability was evaluated [26]. Pollen was collected from the WT and *ms* plants, stained with 1% aceto-carmine, and examined using a microscope (BX51, Olympus Corporation, Tokyo, Japan). Based on crossing of the male sterile mutants *ms-30* and *ms-33* with Heinz 1706, we collected the fruit set data.

### 4.3. Molecular Marker Development and Genotyping

InDels and SNPs were identified by comparing the sequence of chromosome 6 between the tomato lines Heinz 1706 and LA1589. Whole-genome sequences for these lines were obtained from the Sol Genomics Network (SGN, https://solgenomics.net/) [58]. To evaluate polymorphisms between *ms-30*/*ms-33* and LA1589, PCR primers were designed based on the flanking regions of these InDels and SNPs. 

KASP genotyping reactions were performed using a LightCycler480I (Roche Diagnostics GmbH, Mannheim, Germany) with KASP-TF V4.0 2× Master Mix with Low Rox (Cat. No. KBS-1050-122, LGC Genomic Limited, Middlesex, United Kingdom) following the provided manual protocol (https://www.biosearchtech.com/support/education/kasp-genotyping-reagents/running-kasp-genotyping-reactions). The *TPI* gene fragment for *ms-33* was amplified using the marker MS33SEQ and sequenced at the Beijing Genomics Institute (Beijing, China). To identify the insertion of the LTR retrotransposon in the *7B-1* and *sl-2* mutants, PCR was conducted separately with primers LeGLO2-6F/R and SL-2TE-F/R. The resulting PCR products were then combined in a single tube and analyzed through electrophoresis.

### 4.4. Genetic Analysis and Fine-Mapping

Genetic analysis and preliminary mapping were performed using 200 plants and 94 male sterile plants selected from each F_2_ population. For fine-mapping of the *ms-30* and *ms-33* loci, all plants in the F_2_ populations were used. The goodness-of-fit to a 3:1 segregation ratio at the *ms-30* and *ms-33* loci was tested using the chi-square test with 200 individuals from each F_2_ population. For preliminary mapping of the *ms-30* and *ms-33* loci, the first 94 sterile plants in each F_2_ population were genotyped with 2 and 6 InDel markers, respectively. For fine-mapping of the *ms-30* and *ms-33* loci, 76 and 39 recombinants were selected from each F_2_ population using markers HP149 and HP3661, respectively. These recombinants were identified as homozygous for either the *ms-30* or *ms-33* mutant allele and for only one of these two markers. The recombinants were further genotyped with additional markers (Appendix A), and their fertility/sterility was assessed by examining the flowers on at least three inflorescences per plant. 

### 4.5. Gene Prediction and Sequence Polymorphism Analysis

The putative genes in the fine-mapped region containing the *ms*-*30* and *ms*-*33* loci were identified using the tomato gene model (ITAG release 4.0) [37] in the SGN (https://solgenomics.net). 

The genomic sequence of *TPI* from the tomato lines WT-30, *ms-30*, WT-33, and *ms-33* was obtained using overlapping PCR. The last exon and part of the 3′ UTR region of the *TPI* gene in *7B-1* and *sl-2* were amplified by PCR with an elongation time of 7 min, using primers LeGLO2-6F/R [25], and 2× Phanta Max Master Mix (Dye Plus) (Cat. No. P525-03, Vazyme, Nanjing, China). The coding sequences (CDS) of the *TPI* gene were obtained through reverse-transcription PCR (RT-PCR). Fragments were amplified using primers LeGLO2-6F/R, and the CDS of *TPI* were cloned into the *pEASY*-Blunt Zero Cloning Vector (*pEASY*-Blunt Zero Cloning Kit, Cat. No. CB501-02; TransGen Biotech, Beijing, China). Both the amplified fragments and plasmids were sequenced at the Beijing Genomics Institute (Beijing, China). All primers utilized in this study are detailed in Appendix A. Genomic DNA of *7B-1* was provided by Professor Huolin Shen from the College of Horticulture, China Agricultural University.

### 4.6. RNA Extraction, cDNA Synthesis, and qRT-PCR Analysis

The sepals, petals, stamens, and carpels of young flower buds at the meiotic and tetrad stages (3–5 mm in length) [40] were collected from WT-30, *ms-30*, WT-33, *ms-33*, AC, *tpicr-e1*, and *tpicr-e7* plants and then immediately frozen in liquid nitrogen. Each tissue type included three biological replicates, with each replicate comprising samples from at least three plants. Total RNA extraction, cDNA synthesis, and qRT-PCR analysis were conducted following the methods previously described by Cao et al. (2019) [26] and Liu et al. (2019) [10]. All primers used for these analyses are detailed in Appendix A.

### 4.7. RNA-Seq Analysis

A total of 24 RNA-Seq libraries, constructed with RNA from WT-30 and *ms-30*, were sequenced using the Illumina NovaSeq6000 platform at Berry Genomics Corporation (Beijing, China). The clean reads obtained were aligned and quantified against the tomato genome (ITAG4.0) using HISAT2 version 2.1.0 [59] and StringTie version 2.0.6 [60]. Differential gene expression analysis was conducted using the DESeq2 package, employing the MA-plot-based method in R version 3.0.3. *p* values were adjusted using the Benjamini–Hochberg procedure to control the false discovery rate [61]. The fold change for the sepals of WT-30 (SE_WT-30) versus sepals of *ms-30* (SE_*ms-30*), petals of WT-30 (PE_WT-30) versus petals of *ms-30* (PE_*ms-30*), stamens of WT-30 (ST_WT-30) versus stamens of *ms-30* (ST_*ms-30*), and carpels of WT-30 (CA_WT-30) versus carpels of *ms-30* (CA_*ms-30*) was determined using FPKM (fragments per kilobase of transcript sequence per million base pairs sequenced) values. The criteria for identifying DEGs were set as |log2FoldChange| ≥ 1 and the adjusted *p* value < 0.05. GO analysis of the DEGs was performed using their closest homologs in *Arabidopsis* via DAVID (Database for Annotation, Visualization, and Integrated Discovery, https://david.ncifcrf.gov/). The expression levels of certain DEGs, previously reported to be involved in the development of stamens or carpels, were validated through qRT-PCR. 

### 4.8. Construction of Gene Editing Vectors, Plant Transformation, and Selection of Mutant Alleles 

For the CRISPR/Cas9 constructs, two sgRNA binding sites per vector were identified using CRISPR-GE [62] (http://skl.scau.edu.cn/). Primers incorporating sgRNAs and *Bsa* I recognition sites were used to amplify the sgRNA(1)_SlU6-2t_SlU3-5p_sgRNA(2) fragments, with the pCBC-S1 vector as a template. The PCR fragments were then assembled into the binary vector pMGET through the Golden Gate cloning method, following the protocols described by Xing et al. (2014) [63] and Yang et al. (2023) [64]. These constructs were introduced into AC tomato plants via *Agrobacterium tumefaciens* (GV3101)-mediated cotyledon explant transformation, adhering to the method outlined by Yang et al. (2023) [64]. To confirm whether transgenic plants were successfully generated, PCR was performed using primers Cas9F and Cas9R. To identify specific types of edits, two primer pairs, TPI1 and TPI2, were used to amplify sequences including the targets of vector1 and vector2, respectively. The resulting PCR fragments were cloned into the *pEASY*-Blunt Zero Cloning Vector. At least 12 clones per PCR product were sequenced at the Beijing Genomics Institute. Two heterozygous T_0_ plants, each with a 4 bp or 66 bp deletion, were self-pollinated to produce T_1_ generation seeds. Homozygous mutants from the T_1_ generation, lacking CAS9, were identified using primers TPI1, TPI2, and Cas9. All primers used are detailed in Appendix A.

### 4.9. Data Statistical Analysis

The mean values of floral organs length and relative expression were analyzed using IBM SPSS Statistics 20.0 software, with a t-test employed to assess significant differences among the means. The chi-square goodness-of-fit test was conducted by online data analysis platform SPSSPRO (https://www.spsspro.com).

## 5. Conclusions

Overall, this study demonstrated that the *TPI* gene (*Solyc06g059970*) is the causal factor for the phenotypic abnormalities observed in the mutants *ms-30*, *ms-33*, *7B-1*, and *sl-2*. Sequence alterations in *TPI* across these mutants included an SNP each in *ms-30* and *ms-33* and an LTR retrotransposon insertion in both *7B-1* and *sl-2*. Leveraging these sequence variations, three codominant markers—MS30KASP, MS33SEQ, and LeGLO2-6/SL-2TE—were developed. RNA-seq and qRT-PCR analyses indicated that loss-of-function mutations in *TPI* altered the expression of genes involved in stamen and carpel development. These findings may be useful for the rapid development of male sterile lines through molecular marker-assisted backcrossing or CRISPR/Cas9-mediated mutagenesis targeting *TPI*, contributing to hybrid seed production and laying the groundwork for further functional analysis of the *TPI* gene.

## Figures and Tables

**Figure 1 ijms-25-03331-f001:**
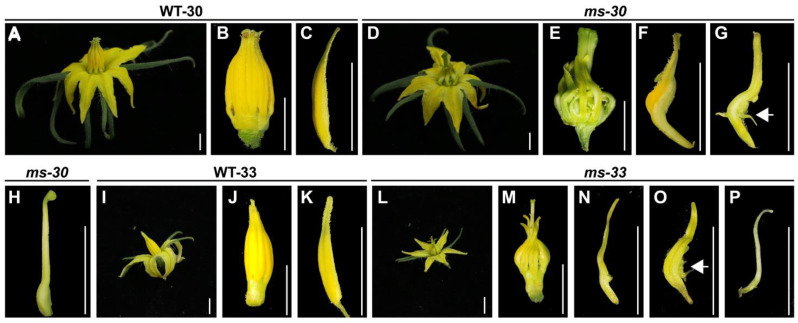
Phenotypes of flowers and stamens in *ms*-*30* and *ms*-*33*. (**A**) Flower, (**B**) anther cone, and (**C**) dissected stamen in WT-30. (**D**) Flower and (**E**) anther cone in *ms*-*30*. (**F**) Dissected stamen with carpelloid structures in *ms*-*30*. (**G**) Dissected stamen with carpelloid structures and external ovules in *ms*-*30*; white arrow denotes the external ovules. (**H**) Completely carpelloid stamen in *ms*-*30*. (**I**) Flower, (**J**) anther cone, (**K**) and dissected stamen in WT-33. (**L**) Flower and (**M**) anther cone in *ms*-*33*. (**N**) Dissected stamen with carpelloid structures in *ms*-*33*. (**O**) Dissected stamen with carpelloid structures and external ovules in *ms*-*33*; white arrow indicates the external ovules. (**P**) Completely carpelloid stamen in *ms*-*33*. The white bar indicates 0.5 cm.

**Figure 2 ijms-25-03331-f002:**
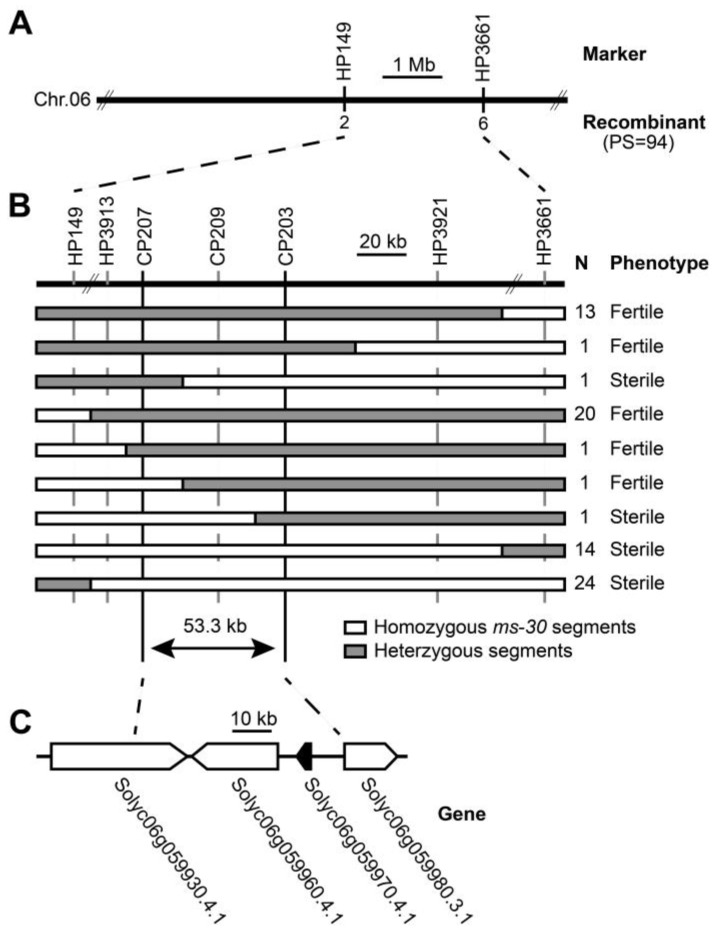
Fine-mapping of *ms*-*30*. (**A**) Preliminary mapping of *ms*-*30*; PS indicates population size. (**B**) Fine-mapping of *ms*-*30*; N indicates the number of recombinants, and 76 recombinants were selected from 949 plants of F_2_ population. (**C**) ITAG4.0 annotated genes of *ms*-*30*. Arrows indicate the direction of transcription, and the solid arrow represents the most likely candidate gene for *ms*-*30*.

**Figure 3 ijms-25-03331-f003:**
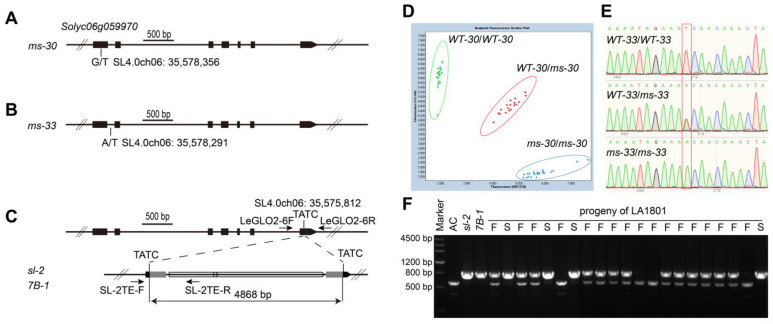
Sequence alterations of the *TPI* gene in *ms*-*30*, *ms*-*33*, *7B*-*1*, and *sl*-*2* and the development of molecular markers (**A**,**B**) Single-nucleotide polymorphisms (SNPs) in the *TPI* gene of *ms*-*30* and *ms*-*33*; black boxes indicate the exon of *TPI*; (**C**) structure of the *TPI* gene according to ITAG4.0 (upper) and the structure of the long terminal repeat (LTR) retrotransposon inserted in the last exon of *TPI* in *7B*-*1* and *sl*-*2* (lower); gray boxes indicate the LTR sequence of retrotransposon, hollow blocks are two ORFs predicted on the retrotransposon, and arrows indicate the direction of transcription. “TATC” is the TSD sequences. LeGLO2-6F, LeGLO2-6R, SL-2TE-F, and SL-2TE-R are the primers used in generating the LeGLO2-6/SL-2TE marker. (**D**) Endpoint fluorescence scatter plot of the Kompetitive Allele Specific PCR marker MS30KASP. (**E**) The chromatogram of MS33SEQ; red box denotes the SNP in *TPI* of *ms*-*33*. (**F**) Agarose gel electrophoresis of PCR fragments amplified from Ailsa Craig, *7B*-*1*, and plants from LA1801 (*sl*-*2*) using the marker LeGLO2-6/SL-2TE; F indicates the fertile plant, whereas S indicates the sterile plant.

**Figure 4 ijms-25-03331-f004:**
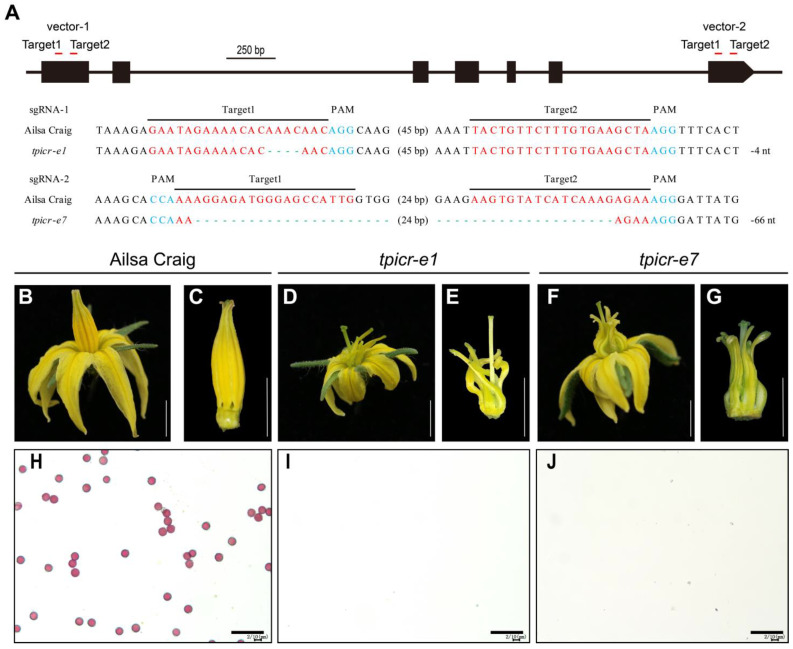
Sequence alterations and phenotypes of *TPI* gene-edited mutants. (**A**) *TPI* mutations generated through CRISPR/Cas9 gene editing. Red letters mark the sequence of targets. Blue letters indicate the protospacer-adjacent motif. Green hyphens indicate deletions in *tpicr* mutants. (**B**–**J**) Representative flower, anther cone, and pollen viability images of AC, *tpicr*-*e1*, and *tpicr*-*e7*. The white bar indicates 0.5 cm (**B**–**G**), and the black bar indicates 1 mm (**H**–**J**).

**Figure 5 ijms-25-03331-f005:**
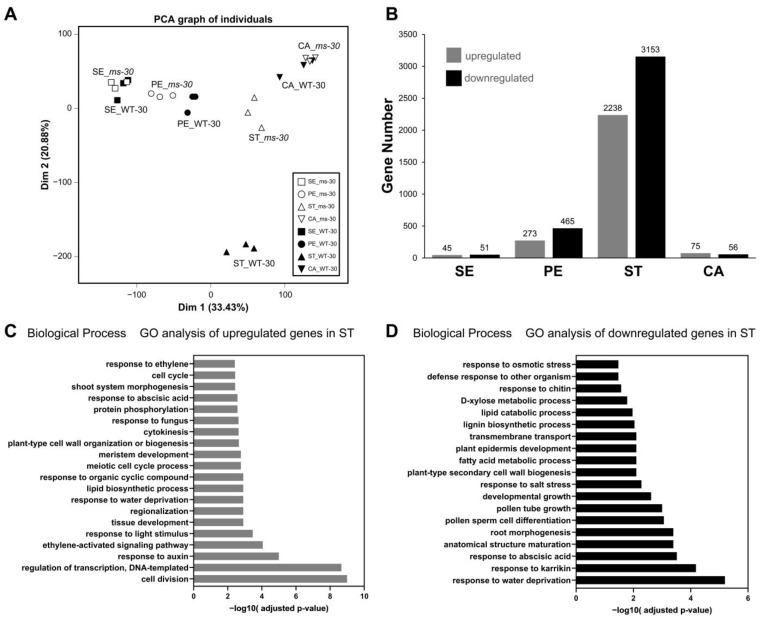
Loss of function of *TPI* in *ms*-*30* affects multiple biological processes during stamen development. (**A**) Principal component analysis of RNA-seq data. (**B**) The number of differentially expressed genes that were upregulated and downregulated in SE, PE, ST, and CA. (**C**) Gene ontology (GO) enrichment analysis of upregulated differentially expressed genes in ST. (**D**) GO enrichment analysis of downregulated differentially expressed genes in ST. SE, Sepals; PE, Petals; ST, Stamens; and CA, Carpels.

**Figure 6 ijms-25-03331-f006:**
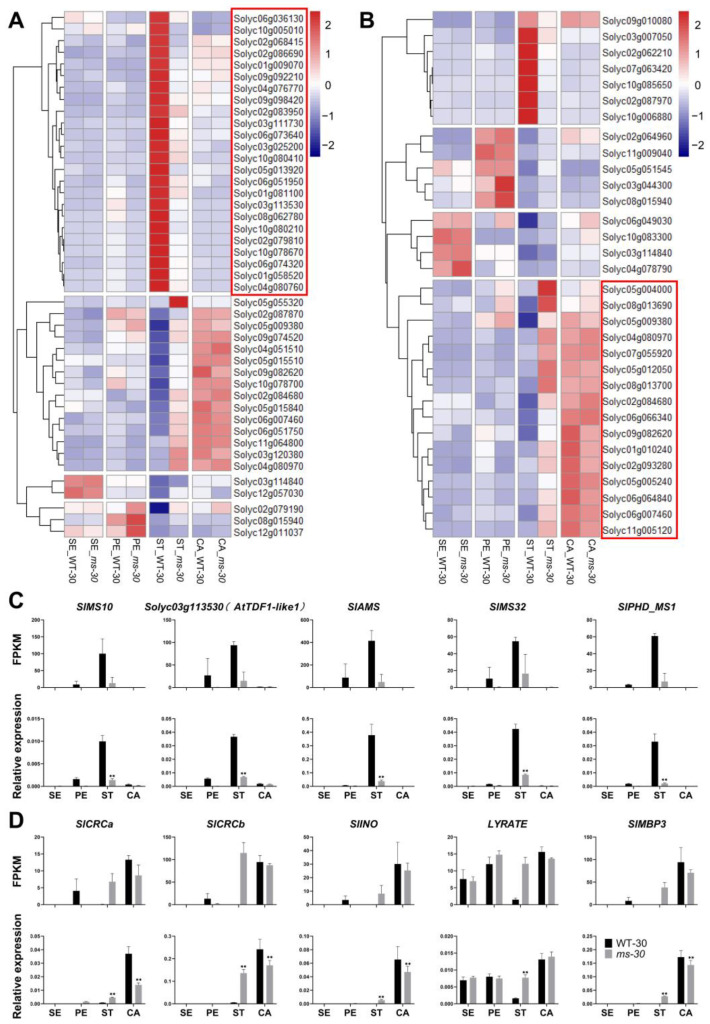
Expression pattern of the DEGs related to stamen or carpel development between WT-30 and *ms*-*30*. (**A**) Heatmap of the DEGs whose homologs in *Arabidopsis* are related to stamen development. The genes in the red box represent those expressed mainly in the stamen of WT-30 and significantly downregulated in the stamen of *ms*-*30*. (**B**) Heatmap of DEGs whose homologs in *Arabidopsis* are related to carpel development. Genes in the red box represent those mainly expressed in the carpel and significantly upregulated in the stamen of *ms*-*30*. (**C**,**D**) FPKM and qRT-PCR analysis of the DEGs related to stamen and carpel development in four floral organs of *ms*-*30*. Asterisks indicate a significant difference (**, *p* < 0.01) between WT-30 and *ms*-*30*; WT-30 and *ms*-*30* are homozygous male fertile plants and male sterile plants derived from the tomato line 2-455, respectively. SE, sepals; PE, petals; ST, stamens; and CA, carpels.

**Table 1 ijms-25-03331-t001:** Chi-square goodness-of-fit test for segregation ratios of F_2_ populations at *ms*-*30* and *ms*-*33* loci.

Locus	Phenotype	Observed Frequency	Expected Frequency	Observed Proportion	Expected Proportion	Residual	χ^2^	*p*
*ms*-*30*	Fertile	139	150	0.695	0.75	−11	3.227	0.072
	Sterile	61	50	0.305	0.25	11		
*ms*-*33*	Fertile	149	150	0.745	0.75	−1	0.027	0.870
	Sterile	51	50	0.255	0.25	1		

**Table 2 ijms-25-03331-t002:** Predicted genes in the *ms*-*30* region.

Gene Name	Position on SL4.0ch06	Putative Function
*Solyc06g059930*	35503650..35543738(+)	sesquiterpene synthase 1
*Solyc06g059960*	35545393..35569077(−)	Histone-lysine N-methyltransferase ASHH2
** *Solyc06g059970* **	**35575492..35578559(−)**	***Tomato PISTILLATA* (*TPI*, syn. *SlGLO2*)**
*Solyc06g059980*	35588500..35603986(+)	O-fucosyltransferase family protein

## Data Availability

The RNA sequencing datasets generated in this study have been deposited in the Sequence Read Archive (SRA) under the accession number PRJNA1036293.

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
