# Peer review of "Candidate Gene Identification and Transcriptome Analysis of Tomato male sterile-30 and Functional Marker Development for ms-30 and Its Alleles, ms-33, 7B-1, and stamenless-2"

_ijms, 2024, doi:10.3390/ijms25063331_

Round 1
Reviewer 1 Report
Comments and Suggestions for Authors
Overall, the results, discussion and methodology section are well framed and succinct.
The authors have done a great job with the study, analysis, interpretation and writing and the paper can be accepted with minor revisions.
The minor changes needed in the paper are:
Abstract:
In the abstract, authors have introduced 7B-1 and SL-2, without explaining what these are. A brief explanation should be provided about these before introducing this term in the abstract.
Introduction:
Again, explanation for 7B-1 is missing and should be provided.
Adding the objective of the study at the end of introduction can enhance paper quality.
The last paragraph at the end of introduction is mostly the compilation of results and needs to be removed from the introduction section. It is more suitable for discussion section.
Reviewer 2 Report
Comments and Suggestions for Authors
The study revealed that the TPI gene is a causative factor in the phenotypic abnormalities observed in the ms-30, ms-33, 7B-1, and sl-2 mutants. These results can be used to create male sterile lines, which is important for the production of hybrid seeds. Some comments.
In the Introduction section, indicate the specific purpose of the study.
In the Materials and Methods section, it is necessary to include a subsection of statistical analysis.
In the Materials and Methods section, give a brief description of the methods used to study flower morphology and pollen viability.
Give the % yield when obtaining transgenic plants. It is known that the cellular response to cell division can include immediate and delayed mechanisms, as shown by the analysis of DEGs in stamens?
What is the mechanism of molecular regulation, according to the authors, of the formation of carpelloid stamens in mutants?
It is necessary to bring the list of references in accordance with the requirements of the journal.
